# Effect of Menaquinone-7 Supplementation on Arterial Stiffness in Chronic Hemodialysis Patients: A Multicenter Randomized Controlled Trial

**DOI:** 10.3390/nu15112422

**Published:** 2023-05-23

**Authors:** Nuanjanthip Naiyarakseree, Jeerath Phannajit, Wichai Naiyarakseree, Nanta Mahatanan, Pagaporn Asavapujanamanee, Sookruetai Lekhyananda, Supat Vanichakarn, Yingyos Avihingsanon, Kearkiat Praditpornsilpa, Somchai Eiam-Ong, Paweena Susantitaphong

**Affiliations:** 1Division of Nephrology, Department of Medicine, Faculty of Medicine, Chulalongkorn University, Bangkok 10330, Thailand; 2Research Unit for Metabolic Bone Disease in CKD Patients, Faculty of Medicine, Chulalongkorn University, Bangkok 10330, Thailand; 3Division of Clinical Epidemiology, Department of Medicine, Faculty of Medicine, Chulalongkorn University, Bangkok 10330, Thailand; 4Division of Nephrology, Department of Medicine, Bangkok Christian Hospital, Bangkok 10500, Thailand; 5Hemodialysis Center BENCHAKITTI–MDCU, Benjakitti Park Hospital, Bangkok 10330, Thailand; 6The Kidney Foundation of Thailand, Bangkok 10700, Thailand

**Keywords:** menaquinone-7, arterial stiffness, carotid-femoral pulse wave velocity, chronic hemodialysis

## Abstract

Background: There is a very high prevalence of subclinical vitamin K deficiency in patients requiring hemodialysis (HD), and this problem is associated with vascular calcification and arterial stiffness. Vitamin K2 (MK-7) supplementation can improve vitamin K status in HD patients. However, the benefits of vitamin K supplementation on arterial stiffness have still not been established. The present study was conducted to evaluate the efficacy of menaquinone-7 (MK-7) supplementation on arterial stiffness in chronic HD patients. Methods: This open-label multicenter randomized clinical trial was conducted in 96 HD patients who had arterial stiffness, defined by high carotid femoral pulse wave velocity (cfPWV ≥ 10 m/s). The patients were randomly assigned to receive oral MK-7 (375 mcg once daily) for 24 weeks (*n* = 50) or standard care (control group; *n* = 46). The change in cfPWV was the primary outcome. Results: Baseline parameters were comparable between the two groups. There was no significant difference in the change in cPWV at 24 weeks between the MK-7 group and standard care [−6.0% (−20.2, 2.3) vs. −6.8% (−19.0, 7.3), *p* = 0.24]. However, we found that MK-7 significantly decreased cPWV in patients with diabetes [−10.0% (−15.9, −0.8) vs. 3.8% (−5.8, 11.6), *p* = 0.008]. In addition, the MK-7 group had a lower rate of arterial stiffness progression, compared to controls (30.2% vs. 39.5%, *p* = 0.37), especially in diabetes patients (21.4% vs. 72.7%, *p* = 0.01). No serious adverse events were observed during the 24 weeks. Conclusion: Vitamin K supplements provided a beneficial impact in lowering the rate of arterial stiffness progression in chronic hemodialysis patients with diabetes. Possible benefits on cardiovascular outcomes require further investigation.

## 1. Introduction

Cardiovascular morbidity and mortality are the leading causes of death in individuals with chronic kidney disease (CKD), particularly those with end-stage kidney disease (ESKD). Nutritional status is also one of the most important factors for survival in dialysis patients. The global prevalence of malnutrition and sarcopenia reported in ESKD patients were 28–54% [1] and 22–29% [2], respectively, which are related to cardiovascular events and mortality [2,3]. Other risk factors associated with cardiovascular events include a chronic inflammatory process driven by uremic toxins and abnormalities in mineral and bone metabolism such as hyperphosphatemia, hypercalcemia, and hyperparathyroidism [4,5]. These factors contribute to arterial stiffness, which refers to structural changes in muscular and elastic vessels that occur prior to atherosclerotic plaque and thrombosis. Arterial stiffness is also considered as a functional marker of arterial calcification. In hemodialysis patients, arterial stiffness is a major contributor to cardiovascular risk, leading to hemodynamic changes through various pathways. Vascular stiffness increases systolic blood pressure (SBP), causes abnormally wide pulse pressure, leads to left ventricular hypertrophy, and reduces coronary perfusion, eventually causing myocardial infarction, heart failure, and cardiovascular death [6,7].

Previous investigations have revealed a positive correlation between arterial stiffness and arterial calcification in patients with chronic hemodialysis (HD), with the severity of the calcification exacerbating arterial stiffness [8]. Arterial calcification is an active process driven by a disbalance between the promoting and inhibiting mechanisms of calcium deposition in the walls of blood vessels. The matrix Gla protein (MGP), which relies on vitamin K for its activity, is a notable inhibitor of calcification. Vitamin K acts as a cofactor in the post-translational γ-carboxylation of MGP. In cases of vitamin K deficiency, MGP cannot activate to fulfill its inhibitory role, resulting in an excess of inactive MGP or dephosphorylated-uncarboxylated MGP (dp-ucMGP) in circulation. This phenomenon indicates a subclinical deficiency of vitamin K and enhances the progression of vascular calcification [9]. There is a very high prevalence of sub-clinical vitamin K deficiency in CKD patients especially in dialysis [10,11]. Inadequate vitamin K intake has been reported in both CKD and ESKD populations, partly due to potassium-rich food restrictions, with dialysis patients displaying a 4.5-fold increase in dp-ucMGP levels, indicating a significant deficiency of vitamin K [11]. Previous studies and recent meta-analyses [12,13] including our previous research on Thai patients with CKD patients [14] have demonstrated that vitamin K deficiency is related to a higher degree of vascular calcification and could serve as an early marker of vascular calcification in patients with CKD and ESKD.

Given that poor vitamin K status predicts cardiovascular disease and mortality in dialysis patients, vitamin K supplementation may improve arterial calcification by activating MGP. Several previous trials have investigated the effect of vitamin K treatment on substantially reducing circulating plasma dp-ucMGP levels in CKD and ESKD patients [15,16,17]. Only a few trials to date have examined the effect of vitamin K supplementation on arterial stiffness in non-dialysis CKD and post-kidney transplant patients [18,19,20,21]. A recent trial did not find significant improvement in arterial stiffness in patients with CKD stage 3B-4 who received oral MK-7 400 mcg/day for 12 months compared with the placebo group [18]. However, the benefits of oral MK-7 supplementation in high-risk patients such as dialysis patients have not yet been explored. This study aimed to investigate the efficacy of vitamin K2 supplementation (Menaquinone-7 375 mcg/day) in reducing the progression of arterial stiffness, as measured by carotid femoral pulse wave velocity (cfPWV), in chronic HD patients with arterial stiffness compared to controls.

## 2. Materials and Methods

### 2.1. Study Design

We conducted an open label, multicenter randomized controlled study between September 2021 until August 2022. All of the participants were recruited from four HD centers in Bangkok, Thailand (King Chulalongkorn Memorial Hospital, The Kidney Foundation of Thailand HD center, Benjakitti Park Hospital, and Bangkok Christian Hospital). The study protocol was approved by the Chulalongkorn University Institutional Review Board (IRB no. 298/64) and registered with the Thai Clinical Trial Registry (TCTR20230217001). Written informed consent was obtained from all participants before enrollment.

### 2.2. Study Population

Subjects were consecutively enrolled using the following inclusion criteria: (1) age ≥ 18 years at screening; (2) undergoing maintenance HD at least two times a week for ≥1 months prior to screening and having shown compliance with HD treatments; (3) cfPWV measurement ≥ 10 m/s at screening visit. The exclusion criteria were as follows: (1) pregnancy and breastfeeding; (2) currently receiving vitamin K antagonist or warfarin as a regular medication; (3) currently receiving and/or history of vitamin K supplementation within three months before screening; (4) history of percutaneous coronary intervention and/or coronary artery bypass grafting, since these patients may suffer from arterial injuries due to puncture or artery harvesting, which might affect the test results of the automatic device; (5) patients with conditions that prevented assessment cfPWV including deformity of peripheral limbs, amputation, and severe peripheral vascular disease.

### 2.3. Intervention and Control Group

Eligible patients were randomly assigned in a 1:1 ratio by computer-generated block of four randomization and were stratified according to underlying diabetes mellitus (diabetes vs. non-diabetes). Patients in the treatment group received oral vitamin K2 (Menaquinone-7, MK-7) 375 mcg/day once daily for 24 weeks. Patients were asked to take five tablets (75 mcg/tablet) at nighttime and apart from phosphate binder intake at least 3 h. The control group received standard treatment without vitamin K supplement. MK-7 was provided by Pharmanord SAE Phamaceuticals (Bangkok, Thailand). The dose of MK-7 was selected based on a previous study suggesting that vitamin K2 has dose dependent effects on uncarboxylated MGP in chronic HD patients at least up to level 360 mcg/day [17]. The participant’s drug compliance was reviewed by investigators and HD nurses at each HD center every four weeks by counting the returned pills.

### 2.4. Outcomes

The primary outcome was change in cfPWV over 24 weeks after randomization. We measured cfPWV using SphygmoCor XCEL Model EM4C (AtCor Medical Inc., Sydney, Australia). cfPWV was performed in the supine position after at least 15 min rest, before each patient’s HD session. The measurements were carried out at 12 and 24 weeks after randomization by a single well-trained operator throughout the study. Increased arterial stiffness was defined by cfPWV ≥ 10 m/s which was proposed in the 2018 ESH/ESC hypertension guidelines [22].

Secondary outcomes were the proportion of patients with progression of arterial stiffness at the end of the study defined by increased cfPWV compared to baseline, change in biochemistry parameters including fasting blood sugar (FBS), hemoglobin (Hb) A1C, blood urea nitrogen (BUN), serum creatinine, calcium, phosphate, albumin, intact parathyroid hormone (iPTH) and lipid profile at baseline, 12, and 24 weeks after randomization and incidence of adverse effects from MK-7. All blood samples were collected for assessment under fasting conditions before the dialysis sessions.

### 2.5. HD Prescription and Standard Medication in Dialysis Units

All of the patients underwent conventional HD using high-flux dialyzers at a blood flow rate of 250–400 mL/min and a dialysate flow rate of 500–800 mL/min. HD frequency was at least twice weekly, with 4 h per session. Dialysate calcium concentration was determined at 2.5–3.5 mEq/L. All of the patients maintained single pool Kt/V values ≥ 1.2 per HD session or standard kt/V ≥ 2.1 in thrice-weekly HD. All patients were advised to continue their regular medications, such as antihypertensive drugs, phosphate binders, and lipid-lowering agents during the study period. Important medications that might contribute to a change in arterial stiffness such as phosphate binders, vitamin D, and calcimimetic drugs, were not allowed to be adjusted throughout the study period.

### 2.6. Statistical Analysis

The sample size calculation was based on previous data from kidney transplant patients due to the absence of a previous trial in the HD population [19]. This study required a total of 96 participants to detect a 14.2% difference in cfPWV reduction with 80% power and 5% type I error rate while accounting for 20% of dropouts.

Continuous variables with normal distribution were expressed as a mean ± standard deviation (SD). In a skewed distribution, the data were presented as median with interquartile range (IQR). Baseline comparisons between groups were obtained using a chi-square test for categorical data and a student’s *t*-test or Wilcoxon rank-sum test for normal and skewed continuous data, respectively.

The intention-to-treat (ITT) principle was applied in all analyses including all participants who were randomized and had undergone at least one follow-up cfPWV measurement. Multivariable linear mixed effect model (LMM) adjusted by baseline cfPWV with treatment and continuous time interaction-term were used to analyze the primary outcome.

Sensitivity analyses for primary outcomes were performed by including only patients who completed a 24-week follow-up and by including blood-pressures and smoking status as adjusting factor using the same LMM modeling.

For secondary outcomes, the chi-square test was used to determine a significant difference between the proportion of participants with progression of arterial stiffness at the end of the study. LMM with continuous-time interaction and treatment were used for all laboratory data. Subgroup analyses were explored for diabetes status, HD frequency, and baseline serum calcium level. A post-hoc analysis to determine predictors of arterial stiffness was performed using multivariable logistic regression.

All reported *p*-values are two-tailed and *p* < 0.05 was statistically significant. Adjustments for multiple comparisons was not performed for all secondary and subgroup analyses. Therefore, the *p*-value should be interpreted with caution. All statistical analyses were performed with STATA^®^ (StataCorp. Version 16. College Station, TX, USA).

## 3. Results

Among the 144 HD patients who met the inclusion and exclusion criteria, 96 patients gave their consent to participate in the study. After the enrollment was finished, there were 50 patients in the treatment group (oral MK-7 375 mcg/day) and 46 patients in the control group as shown in Figure 1. A total of 7 patients in the treatment group and 3 patients in the control group discontinued the study before the end of the 24-week study period. The reasons for discontinuation in the treatment group were due to coronavirus disease 2019 (COVID-19) infection (*n* = 2), receiving kidney transplant (*n* = 2), mild gastrointestinal effects (*n* = 1), arteriovenous fistula (AVF) thrombosis, refusal to continue in the trial (*n* = 1), and receiving warfarin due to paroxysmal atrial fibrillation (AF) (*n* = 1). The reasons for the discontinuation in the control group were moving to another HD center (*n* = 1), receiving warfarin due to AVF thrombosis (*n* = 1), and having recurrent colon cancer requiring chemotherapy (*n* = 1).

### 3.1. Baseline Characteristics

There were no significant differences in baseline demographics, comorbidities, current medication, dialysis factors, and laboratory values between the study groups, except for the significant number of patients in the control group who had a history of kidney transplants. All patients received an adequate dose of dialysis during baseline measurements (Table 1). The baseline cfPWV did not differ between the study groups (Figure 2).

### 3.2. Primary Outcome

During the study period, 89 patients underwent follow-up measurements and were included in the ITT population and 86 patients completed follow-up at week 24. There was no significant difference in mean cfPWV between groups [oral MK-7; 12.2 (11.0, 14.3) m/s to 11.7 (10.2, 14.2) m/s vs. control; 12.1 (11.0, 13.4) m/s to 11.4 (9.8, 13.1) m/s, *p* = 0.24] (Figure 2). The oral MK-7 group did not show significant improvement in the absolute cfPWV change [treatment −0.9 (−3.0, 0.3) m/s vs. −0.8 (−2.6, 0.8) m/s, *p* = 0.39] and percent cfPWV change in comparison to controls [treatment −6.0% (−20, 2.3) vs. control −6.8% (−19, 7.3), *p* = 0.51]. Sensitivity analyses were performed in patients who completed follow-up. Adjusting for smoking and blood pressure, this group also showed a non-significant difference in absolute and percent cfPWV change (Table 2, and Appendix A).

### 3.3. Secondary Outcomes

The proportion of the treatment group with an increased progression of cPWV was lower than the control group as shown in Table 2 (oral MK-7 30.2% vs. control 39.5%, *p* = 0.37). After 12 and 24 weeks of treatment, laboratory parameters of metabolic and bone turnover changes from baseline did not reveal significant differences between groups, except that serum albumin was lower in the control group (*p* = 0.03) (Appendix A).

### 3.4. Subgroup Analyses

The pre-specified subgroup analyses are illustrated in Table 2. DM patients in the treatment group showed a significant improvement in absolute cfPWV change [treatment −0.7 (−2.5, −0.1) m/s versus control 1.3 (0.0, 2.2) m/s, *p* = 0.012] and percent change in cfPWV [treatment −5.1% (−16.1, −0.6) vs. control 8.2% (0.0, 17.2), *p* = 0.01] compared to the control group. Furthermore, DM patients in the treatment group had significantly less progression of cfPWV than control (treatment 21.4% vs. control 72.7%, *p* = 0.01) (Table 2, Figure 2b, and Appendix A).

In patients without DM, there was no significant differences in the absolute cfPWV change and percent cfPWV change in both groups at 12 weeks and 24 weeks. Both oral MK-7 treatment and control groups exhibited similar progression of cfPWV (treatment 34.5% vs. control 28.1%, *p* = 0.59) (Table 2, Figure 2c, Appendix A).

In the subgroup, there were no significant differences in the percent change of cPWV and cPWV progression in both baseline serum Ca > 10 mg/dL and serum Ca ≤ 10 mg/dL (Table 2). 

HD frequency was also explored by subgroup analysis, patients who received oral MK-7 treatment group have a less percent change in cPWV at 24 weeks compared with the control group in the thrice-a-week HD subgroup [−5.8 (−19.6, 2.3)% vs. −10.0 (−21.7, 2.8)%, *p* = 0.024]. But no significant difference between the treatment and control was shown in cPWV progression of the subgroup HD 3 times/week [31.7% vs. 33.3%), *p* = 0.88]. However, there was no significant change in the percent change in cPWV and cPWV progression between treatment and control at 24 weeks in subgroup with HD frequency 2 times/week. [0% vs. 71.4%, *p* = 0.07) (Table 2).

Post-hoc analysis using univariate and multivariable logistic regression that included intervention, age, diabetes status, SBP, HD frequency, dialysate Ca concentration, and smoking status did not find any independent predictors of the progression of arterial stiffness (Appendix A).

### 3.5. Adverse Events and Compliance

During the 24-week follow-up period, adverse events were reported in 9 participants (Table 3). Three patients in the treatment group experienced nausea and abdominal discomfort after taking oral MK-7 within the first two weeks. Their symptoms were treated with domperidone, and two patients were able to continue treatment without further problems. One patient withdrew from the study due to an intolerance to gastrointestinal side effects and one patient in the treatment group developed AF during a session of HD and was subsequently started on warfarin for clot prophylaxis. This patient dropped out of the study after 10 weeks of treatment. Two patients, one in each group, developed AVF thrombosis, the patient in the control group had a history of multiple episodes before randomization. This patient was started on warfarin for secondary prophylaxis after vascular intervention at 12 weeks. The patient in the treatment group who developed AVF thrombosis had no history of thrombosis or stenosis before, and after successful AVF thrombectomy, refused to continue in the study at 20 weeks of treatment. One patient was hospitalized for COVID-19 and 1 patient was diagnosed with colon cancer after randomization and was excluded from the study. There was no mortality reported during the study period. Other participants showed good tolerance to MK-7 and reported good compliance (>95% by counting the returned pills).

## 4. Discussion

This study represents the first RCT to investigate the impact of vitamin K2 supplementation on vascular stiffness in ESKD patients who had high risk of cardiovascular disease due to severe arterial stiffness as determined by PWV. Following a 24-week intervention period, no significant difference in cfPWV between groups was demonstrated. However, the oral MK-7 group showed a greater reduction in cfPWV among patients with diabetes compared with the control group. The MK-7 group did exhibit an ability to slow the progression of arterial stiffness, which was more pronounced in patients with diabetes.

Vitamin K serves as a crucial cofactor for MGP in inhibiting vascular calcification. The main objective of vitamin K supplementation is to alleviate subclinical vitamin K deficiency and inhibit the progression of vascular calcification in patients with CKD and ESKD. Vitamin K supplementation has been shown to reduce plasma dp-ucMGP levels, indicating an improvement in vitamin K deficiency in CKD patients [15,16,23]. More than 90% of hemodialysis patients have been diagnosed as subclinical vitamin K deficiency [10,24]. Previous studies have used oral MK-7 at a dose range of 360 to 400 mcg/day for at least six weeks to reduce plasma levels of dp-ucMGP in CKD and dialysis patients [15,16,19]. However, there has been controversy regarding the potential of vitamin K to retard arterial stiffness or vascular calcification in patients with CKD and ESKD patients. A recent randomized placebo-controlled trial revealed that vitamin K2 supplementation (oral MK-7 at 400 mcg/day) did not improve arterial stiffness in pre-dialysis patients with CKD stage 3b-4 over a 12-month period. Several limitations and explanations were proposed, including the lower incidence of vitamin K deficiency and lower risk of arterial stiffness in this CKD population as well as the short follow-up period [18]. However, ESKD patients, such as those in our study population, who have a very high risk may derive benefits. Our study prescribed a daily dose of 375 mcg of MK-7 for at least 24 weeks and revealed a trend towards slowing progression of vascular stiffness, with a prominent effect observed in ESKD patients with diabetes mellitus. This effect may stem from the direct impact of vitamin K supplementation, as our follow-up analyses did not indicate any alterations in well-known risk factors such as metabolic and bone turnover parameters (Appendix A). Future research is required to examine the long-term clinical outcomes.

Patients with DM generally have an accelerated risk of developing vascular stiffness, even if they have normal renal function. Previous studies have shown that the increase in patients with the degree of PWV in DM is correlated with higher levels of plasma dp-ucMGP in patients with DM even without CKD [23,25]. In a subgroup analysis of DM patients in our study, we found that vitamin K supplementation could attenuate the progression of arterial stiffness in ESKD patients with DM after 12 weeks of treatment, and the effect remained at 24 weeks of MK-7 treatment. Patients with DM in the treatment group also had less progression of arterial stiffness compared with the control group, despite having slightly higher baseline HbA1C and blood sugar levels. The beneficial effect of vitamin K supplementation was not observed in ESKD patients without DM. A previous study demonstrated that vitamin K may help enhance insulin sensitivity in pancreatic β-cells, thereby lowering plasma glucose levels [26]. Another explanation is that vitamin K is also a cofactor in the activation of osteocalcin, which plays a role in enhancing pancreatic β-cell function, leading to an improvement in insulin sensitivity [27]. Furthermore, osteocalcin increases glucose uptake into human muscle and diminishes hepatic gluconeogenesis via adiponectin. Our study also revealed a trend towards a decrease in FBS and HbA1C in the treatment group, but there was no statistically significant difference between baseline and 24 weeks of treatment (Table 1 and Appendix A). Therefore, our study suggests that vitamin K2 may benefit arterial stiffness dependent on glucose levels. Another mechanism proposed by our hypothesis is that the chronic inflammatory state in DM patients might aggravate insulin resistance via inflammatory mediators such as interleukin-6 (IL-6) and nuclear factor kappa B (NF-kB). Previous studies have demonstrated that vitamin K2 has the potential to suppress these inflammatory mediators in humans [28,29]. However, the beneficial effect of vitamin K2 on inflammatory markers needs further exploration.

Our prior investigation revealed that serum calcium and hemodialysis adequacy were independently associated risk factors with an increase in arterial stiffness in addition to blood sugar [30]. In order to reduce the risk of arterial stiffness, we conducted a subgroup analysis to categorize patients with high serum calcium versus low serum calcium as a critical factor in vascular remodeling, and those undergoing hemodialysis twice per week versus three times per week to alleviate volume overload. Our study did not demonstrate a significant improvement in arterial stiffness regardless of serum calcium. Interestingly, thrice-weekly HD had an effect of retarding cfPWV progression. These findings can be attributed to the multifactorial etiologies of arterial stiffness, which is strongly linked to aging, premature vascular aging, blood pressure control, mineral metabolism disturbances, and accumulation of advanced glycation end-products (AGEs). Despite the accumulation of traditional and dialysis-related cardiovascular risk factors that could partially explain the acceleration of arterial stiffness in chronic HD patients, the underlying mechanisms remain unclear. We suggest the adoption of a multidisciplinary treatment approach to mitigate the progression of vascular stiffness in ESKD patients.

Another noteworthy finding of this study is that MK-7 was well-tolerated by ESKD patients, with only mild adverse effects reported. The most common adverse events observed were gastrointestinal events, including nausea, vomiting, and abdominal discomfort, all of which could be alleviated with supportive treatment with antiemetic medication. Additionally, taking the supplement at bedtime was shown to reduce these effects. Importantly, no fatal thrombotic events were observed after vitamin K supplementation, which is supported by previous reports [15,16,23]. Given the relatively minor adverse effects observed and the significant benefits associated with decreased arterial stiffness and cardiovascular morbidity in ESKD patients, the investigators concluded that vitamin K supplementation is a safe and valuable treatment option.

It is important to acknowledge the limitations of our study. Firstly, the main outcome measured in our study was the change in cfPWV. Although cfPWV has been accepted as a standard tool for the assessment of arterial stiffness [31,32] and has been shown by a recent study to be an independent predictor of all-cause and cardiovascular mortality in patients with chronic HD patients [32], it is still a surrogate endpoint and not a hard clinical outcome. Therefore, the impact of vitamin K supplementation on clinical outcomes should be further explored. Furthermore, the number of participants was relatively small, we calculated the sample size based on a 14.2% cfPWV reduction from previous data [19] which may not have power to detect a smaller difference. Additionally, the follow-up period of our study was relatively short, and a longer observational period would provide enhanced methodological soundness regarding the progression of vascular stiffness and cardiovascular mortality. Another limitation is that the severity of vitamin K deficiency was not explored in our study due to the limitations of the diagnostic criteria of vitamin K deficiency. There were still no established consensus levels for the diagnosis and assessment of the degree of severity of vitamin K deficiency by using the direct vitamin K levels or indirect biomarker measurement of vitamin K status such as serum osteocalcin or serum dp-ucMGP levels. Therefore, we were unable to demonstrate a correlation between the improvement in vitamin K deficiency and the decrease in vascular stiffness in ESKD patients. Our study included participants with adequate dialysis who were non-malnourished, supported by normal serum albumin in both groups, so the results should be generalized to only these group of patients. Lastly, neither statistical adjustment of multiple analyses nor power calculation were completed for secondary outcomes, subgroup and post-hoc analyses. The statistical significance of these findings should be interpreted with caution and should be concluded as hypothesis-generating which require further investigation.

## 5. Conclusions

Vitamin K supplementation was found associated with a lower rate of arterial stiffness progression in chronic hemodialysis patients with diabetes. However, research concerning possible benefits on cardiovascular outcomes are still needed.

## Figures and Tables

**Figure 1 nutrients-15-02422-f001:**
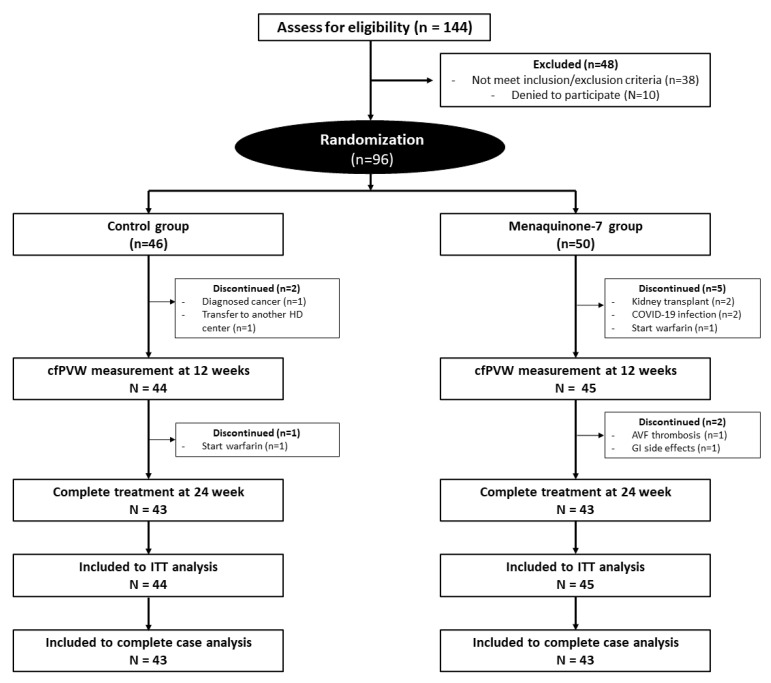
Schematic illustration of study design, randomization, and drop-outs. Abbreviation: cfPVW, carotid femoral pulse wave velocity; ITT, intention-to-treat; COVID-19, Coronavirus disease 2019.

**Figure 2 nutrients-15-02422-f002:**
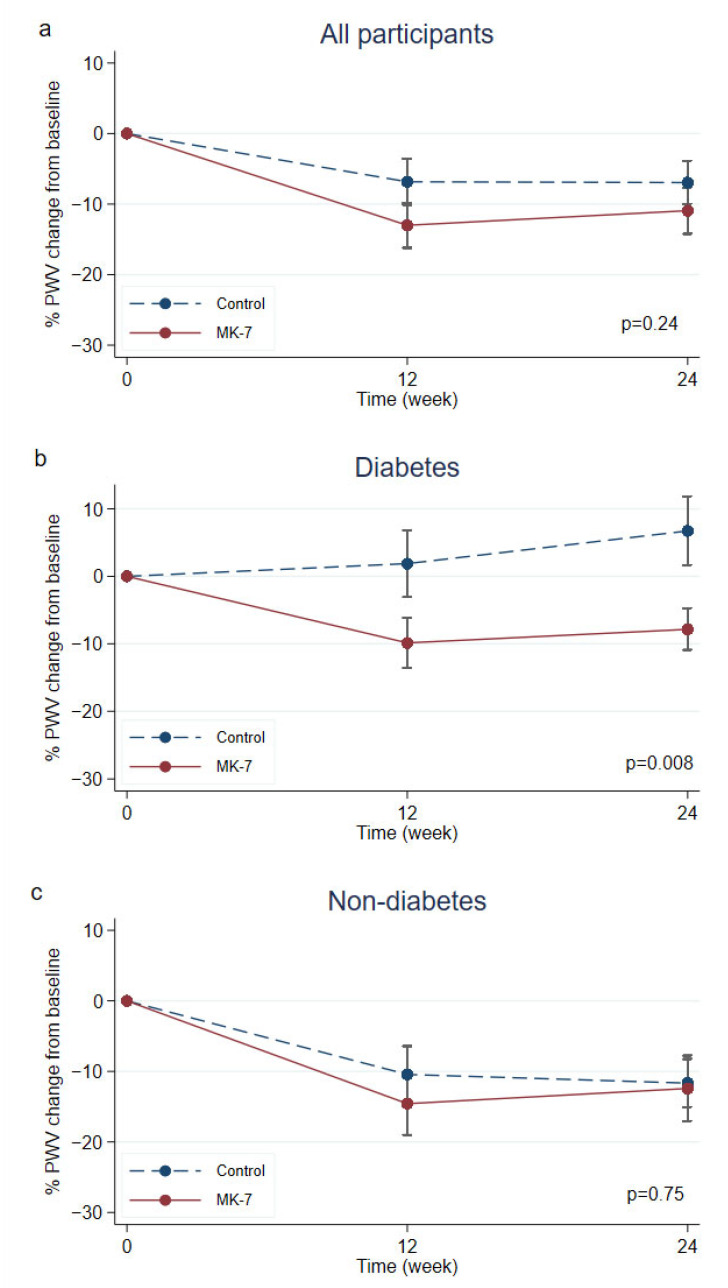
Percentage change of cfPWV from baseline, 12 and 24 weeks in treatment and control groups, (**a**) shows percent change of cfPWV in all participants, (**b**) shows the percentage change of cfPWV in patients in the subgroup with DM, (**c**) shows percent change of cfPWV in subgroup patients without DM.

**Table 1 nutrients-15-02422-t001:** Baseline characteristics of study patients.

Baseline Characteristics	Control Group (*n* = 46)	MK-7 Group (*n* = 50)	*p*-Value
Age (years)	60.0 ± 11.8	59.7 ± 11.3	0.88
Male (*n*/%)	26 (57%)	26 (52%)	0.66
Dry weight (kg)	59.1 ± 13.2	57.1 ± 11.4	0.43
Blood pressure (mmHg)			
Systolic BP	155.5 ± 21.1	148.0 ± 20.2	0.07
Diastolic BP	86.2 ± 13.3	81.5 ± 12.7	0.07
Cause of ESKD (*n*/%)			0.99
Diabetic nephropathy	14 (30%)	17 (34%)	
Hypertension	2 (4%)	3 (6%)	
Glomerulonephritis	12 (26%)	12 (24%)	
Others	3 (7%)	3 (6%)	
Unknown cause	15 (33%)	15 (30%)	
HD vintage (years)	10.2 (3.9–19.2)	6.5 (2.5–17.3)	0.3
Previous kidney transplant (*n*/%)	6 (13%)	1 (2%)	0.03
Underlying diseases (*n*/%)			
Diabetes mellitus	14 (30%)	16 (32%)	0.87
Hypertension	43 (93%)	46 (92%)	0.78
Coronary artery disease	2 (4%)	2 (4%)	0.93
Current smoking (*n*/%)	3 (7%)	0 (0%)	0.06
Antihypertensive drugs no. (*n*/%)			0.88
None	6 (13%)	5 (10%)	
1–4	36 (78%)	41 (82%)	
>4	4 (9%)	4 (8%)	
Antiplatelet drug use (*n*/%)	12 (26%)	14 (28%)	0.83
Lipid lowering drug use (*n*/%)	32 (70%)	37 (74%)	0.63
Phosphate binder no. (*n*/%)			0.29
none	6 (13%)	5 (10%)	
1	27 (59%)	34 (68%)	
2	10 (22%)	11 (22%)	
3	3 (7%)	0 (0%)	
Oral calcium use (*n*/%)	36 (78%)	37 (74%)	0.63
Aluminum hydroxide use (*n*/%)	5 (11%)	5 (10%)	0.89
Lanthanum carbonate use (*n*/%)	9 (20%)	7 (14%)	0.46
Sevelamer use (*n*/%)	6 (13%)	7 (14%)	0.89
Vitamin D3 use (*n*/%)	23 (50%)	25 (50%)	1.00
Calcimimetic drug * use (*n*/%)	4 (9%)	3 (6%)	0.61
HD frequency (*n*/%)			0.08
2 times/week	8 (17%)	3 (6%)	
3 times/week	38 (83%)	47 (94%)	
Dialysate calcium (mEq/L)			0.41
2	2 (4%)	0 (0%)	
2.5	25 (54%)	31 (62%)	
3	9 (20%)	7 (14%)	
3.5	10 (22%)	12 (24%)	
Residual urine (mL/day)	0.0 (0.0–0.0)	0.0 (0.0–0.0)	0.77
HD adequacy (kT/V)	1.9 (1.8–2.1)	2.0 (1.8–2.3)	0.26
Hemoglobin (mg/dL)	10.2 ± 1.7	10.5 ± 1.3	0.39
Baseline cfPWV (m/s)	12.1 (11.0–13.4)	12.2 (11.0–14.3)	0.64
Pre-dialysis BUN (mg/dL)	55.5 ± 13.7	59.0 ± 16.5	0.27
Serum creatinine (mg/dL)	10.2 ± 2.6	9.3 ± 2.3	0.06
Serum calcium (mg/dL)	9.3 ± 0.8	11.2 ± 14.5	0.37
Serum phosphate (mg/dL)	4.3 ± 1.3	4.2 ± 1.4	0.71
Serum albumin (mg/dL)	4.3 ± 0.8	4.1 ± 0.4	0.14
FBS (mg/dL)	97.9 ± 24.8	115.0 ± 63.2	0.09
HbA1C (%)	5.4 ± 0.8	5.9 ± 1.3	0.05
Total cholesterol (mg/dL)	160.5 ± 30.4	167.4 ± 36.4	0.39
Triglyceride (mg/dL)	108.4 ± 57.3	117.1 ± 74.5	0.58
LDL-C (mg/dL)	98.6 ± 39.9	93.2 ± 35.5	0.50
HDL-C (mg/dL)	52.1 ± 15.9	56.6 ± 19.6	0.33
iPTH (pg/mL)	416.6 ± 493.3	473.0 ± 456.7	0.58

Continuous data were shown with mean ± SD or median (interquartile range), categorical data were shown with count (percentage). * all calcimimetics drugs were oral cinacalcet. Abbreviation: MK-7, menaquinone-7; BP, blood pressure; BUN, blood urea nitrogen; FBS, fasting blood sugar; HD, hemodialysis; HDL-C, high density lipoprotein cholesterol; iPTH, intact parathyroid hormone; LDL-C, low density lipoprotein cholesterol.

**Table 2 nutrients-15-02422-t002:** Primary and key secondary outcomes with prespecified subgroup analysis.

	Control Group	MK-7 Group	*p*-Value
Percentage change of cfPWV in 24 weeks (%)			
All patients (ITT population)	N = 44	N = 45	0.24
	−6.7 (−18.8–7.8)	−6.0 (−19.6–2.3)	
Participants with complete 24-week follow-up	N = 43	N = 43	0.24
	−6.8 (−19.0, 7.3)	−6.0 (−20.2, 2.3)	
Diabetes subgroup			
DM	N = 14	N = 16	0.008
	3.8 (−5.8, 11.6)	−10.0 (−15.9, −0.8)	
Non-DM	N = 32	N = 34	0.75
	8.2 (0.0,17.2)	−5.1 (−16.1, −0.6)	
Subgroup divided by baseline serum calcium			
Baseline Ca > 10 mg/dL	N = 36	N = 39	0.73
	0.0 (−17.2, 12.7)	−4.5 (−20.2, 2.6)	
Baseline Ca ≤ 10 mg/dL	N = 10	N = 11	0.14
	−18.8 (−22.7, −12.1)	−8.4 (−16.4, −5.8)	
Subgroup divided by frequency of HD			
HD frequency 2 times/week	N = 8	N = 3	0.71
	13.4 (0.0, 17.2)	−18.6 (−20.2, −17.0)	
HD frequency 3 times/week	N = 38	N = 47	0.024
	−10.0 (−21.7, 2.8)	−5.8 (−19.6, 2.3)	
Patients with progression of arterial stiffness at week 24 *, *n* (%)			
All patients	N = 46	N = 50	0.37
	17 (39.5%)	13 (30.2%)	
Diabetes subgroup			
DM	N = 14	N = 16	0.01
	8 (72.7%)	3 (21.4%)	
Non-DM	N = 32	N = 34	0.59
	9 (28.1%)	10 (34.5%)	
Subgroup divided by baseline serum calcium			
Baseline Ca ≤ 10 mg/dL	N = 36	N = 39	0.27
	16 (48.5%)	12 (35.3%)	
Baseline Ca > 10 mg/dL	N = 10	N = 11	0.94
	1 (10%)	1 (11.1%)	
Subgroup divided by frequency of HD			
HD frequency 2 times/week	N = 8	N = 3	0.07
	5 (71.4%)	0 (0.0%)	
HD frequency 3 times/week	N = 38	N = 47	0.88
	12 (33.3%)	13 (31.7%)	

* defined by increased cfPWV from baseline (*n*/%). Abbreviation: MK-7, menaquinone-7; cfPWV, Carotid-femoral pulse wave velocity; Ca, serum calcium; DM, Diabetes mellitus; HD, Hemodialysis.

**Table 3 nutrients-15-02422-t003:** Adverse events.

	Control Group (*n* = 46)	MK-7 Group (*n* = 50)	*p*-Value
Gastrointestinal side effects (nausea and abdominal discomfort)	0	3 (6%)	0.24
New onset atrial fibrillation	0	1 (2%)	0.9
AVF thrombosis	1 (2.2%)	1 (2%)	0.9
COVID-19 infection	0	2 (4%)	0.49
Diagnosed cancer during study	1 (2.2%)	0	0.48

Abbreviation: MK-7, menaquinone-7; AVF, arteriovenous fistula; COVID-19, Coronavirus disease 2019.

## Data Availability

The data presented in this study are available on request from the corresponding author.

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
