# Peer review of "Effect of Menaquinone-7 Supplementation on Arterial Stiffness in Chronic Hemodialysis Patients: A Multicenter Randomized Controlled Trial"

_nutrients, 2023, doi:10.3390/nu15112422_

Round 1
Reviewer 1 Report
as a nephrologist, I found this paper nice and important
however, major revision should be done in order to improve tihs paper:
first, Extensive editing of English language required
please revise introduction section, highlight the prevalence and importance of manutritionin dialysis patients .
methoids section should be revised and detailed- how did you defined chronic dialysis ? 3 months?
were patients with previous KT or PD were included?
what about dialysis vintage ? were all patients treated at least 2 years????
why so many patients were treated with high ca dialysate?
how did you ensure complience of treatment ?
please revise heading of tables - subheadings are not clear
what about survival and mortality?
adverese events should be presented in a table with comparison
I would also add a regression analysis for assesing predictors of progression of arterial stiffness
Did you assesed correlation between dialysis vintage , arterial stiffness and MKK7 ? did you detect any advantage among malnutrished patients and patients treated with warfarin?
please add discussion section paragraphs regarding study limitations and clinical implication of the study
would longer study would change the results?
Extensive editing of English language required
Author Response
Response to reviewer #1
- As a nephrologist, I found this paper nice and important however, major revision should be done in order to improve this paper: first,
Author’s response:
Thank you very much for you comment. We believe that our findings contribute to the growing body of research on the potential benefits of vitamin K supplementation in hemodialysis patients with arterial stiffness.
- Extensive editing of English language required
Author’s response:
We have sent the revised manuscript for English editing by one of our author (S.E.) who is fluent in English writing.
- please revise introduction section, highlight the prevalence and importance of malnutrition in dialysis patients.
Author’s response:
Thank you for your suggestion, we have revised and added the prevalence and importance of malnutrition to the introduction section. [line 48-50]
- methods section should be revised and detailed- how did you defined chronic dialysis ? 3 months?
Author’s response:
We included participant undergoing maintenance hemodialysis at least 2 times a week for ≥1 months prior to screening [mentioned in Line 106-107]
- were patients with previous KT or PD were included?
Author’s response:
There are 7 previous KT patients and no previous PD patient included in our study.
The numbers were shown in Table 1, page 6
- What about dialysis vintage? were all patients treated at least 2 years????
Author’s response:
The median (IQR) of HD vintage was 10.2 (3.0-19.2) and 6.5 (2.5-17.3) years in control and MK-7 group respectively (shown in Table 1, page 6)
- Why so many patients were treated with high ca dialysate?
Author’s response:
The dialysate Ca concentration were adjusted based on serum calcium and patients’ symptoms by attending nephrologist per standard local guideline to keep normal serum calcium without intra-dialytic complication.
- How did you ensure complience of treatment ?
Author’s response:
Patients’ drug compliance was reviewed by investigators and HD nurses from each HD centers every 4 weeks by counting returned pills. [mention in line 125-127]
We also reported compliance in result section (subsection 3.5; Adverse event and compliance) [line 273-274]
- please revise heading of tables - subheadings are not clear
Author’s response:
Thank you for your suggestion, we try to redesign the table headings to ensure understanding.
- What about survival and mortality?
Author’s response:
There were no mortality reported during the study period (24 weeks).
- adverse events should be presented in a table with comparison.
Author’s response:
Thank your for your suggestion, we presented it in the new Table 3.
- I would also add a regression analysis for assessing predictors of progression of arterial stiffness.
Author’s response:
We performed an additional post-hoc analysis by univariate and multivariable logistic regression including in-tervention, age, diabetes status, SBP, HD frequency, dialysate Ca concentration, and smoking status did not found an independent predictor of the progression of arterial stiffness. [line 253-256 and the supplementary Table S4]
Please remarked that this study might had an inadequate power to define a potential predictor, the statistical significance of these findings should be interpreted with caution and should be concluded as hypothesis generating which required further investigation. [mentioned in limitation line 390-393]
- Did you assesed correlation between dialysis vintage, arterial stiffness and MK7? did you detect any advantage among malnutrished patients and patients treated with warfarin?
Author’s response:
We have investigated the correlated factors of vascular stiffness in our prior investigation in 144 HD patients which considered more adequate powered (ref no 26,27) “The prevalence of increased arterial stiffness was 73.6%. Multivariable analysis showed that older age, less frequent HD, lower HD adequacy, higher serum calcium, and fasting blood glucose were independently associated with arterial stiffness. HD vintage showed non-significant correlation in the univariate analysis.” We mentioned the findings of this study in the discussion section (line 344-346)
The participants in our study were considered to have a good nutritional status by serum albumin were normal (around 4.1-4.3 g/dL) in both groups. However, we did not explore other details like body composition, muscle mass, or current protein intake. So, the advantage among malnourished patients still need further investigation. We also added the limitation regarding this issue (line 388-390)
In the present study, we excluded patients who were treated with warfarin due to potential drug-drug interaction with vitamin K which could be harmful.
- Please add discussion section paragraphs regarding study limitations and clinical implication of the study
Author’s response:
We added limitations in the last paragraph (line 370-393, page 12-13) and clinical implications in the conclusions (page 13)
- Would longer study would change the results?
Author’s response:
As we reviewed previous trial using vascular stiffness outcomes, the follow-up period ranging between 8-24 week (6 months). (ref no. 20, Saengpanit et al, Nephro 2018, DOI: 10.1159/000488009, and Ohira et al. Diabetes Metab Syndr Obes. 2014 Jul 17;7:313-9. doi: 10.2147/DMSO.S65275.) Unlike vascular calcification, vascular stiffness might have a shorter period of response after intervention. However, longer studies with clinically important outcome such as CV events or mortality is suggested. (Mentioned in our limitation line 370-381)
Reviewer 2 Report
This manuscript, entitled ”Effect of Menaquinone-7 Supplementation on Arterial Stiffness in Chronic Hemodialysis Patients: A Multicenter Randomized Controlled Trial” was a prospective multi-center clinical study to investigate the effect of menaquinone-7(MK-7) supplement upon arterial stiffness in one cohort of hemodialysis population. I have a few concerns in the study design and result interpretation. The detailed information is in below.
Major
1. Although authors have stated the participant number choice in the statistical paragraph, the study participants were relative small number and may weaken the statistical power.
2. The tool using to evaluate arterial stiffness was cfPWV. A single tool measurement may be inadequate to reveal arterial stiffness in CKD esp. in dialysis patients.
3. The observational period was relative short, 24 weeks. Thus, a significant finding of arterial stiffness in hemodialysis patients could be not significant changes.
4. No circulating MK-7 levels in the study subjects were found in the text. Therefore, it is difficult to ascertain the compliance of MK-7 supplement in the study subjects.
Minor
1. P-binder category, calcimimetic drugs(oral or iv injection)?
2. The participants had a well-controlled lipid profile, lipid-lowering agents were routinely prescribed in the HD subjects?
3. Circulating MK-7 levels was not sig. different between two groups at 24 weeks. How to explain? Is this one of reasons of non-significant change in arterial stiffness in the study subjects?
4. What was the circulating MK-7 levels in DM vs non-DM subjects?
A few English grammar sentences could be improved.
Author Response
Response to reviewer #2
This manuscript, entitled “Effect of Menaquinone-7 Supplementation on Arterial Stiffness in Chronic Hemodialysis Patients: A Multicenter Randomized Controlled Trial” was a prospective multi-center clinical study to investigate the effect of menaquinone-7(MK-7) supplement upon arterial stiffness in one cohort of hemodialysis population. I have a few concerns in the study design and result interpretation. The detailed information is in below.
Major
- Although authors have stated the participant number choice in the statistical paragraph, the study participants were relative small number and may weaken the statistical power.
Author’s response:
The sample size was calculated based on previous data in kidney transplant patients (reference no.23) with relatively large effect size (14.2%) reduction which resulting in less power to detect a smaller effect. We added this issue to our limitation. [line 376-378]
- The tool using to evaluate arterial stiffness was cfPWV. A single tool measurement may be inadequate to reveal arterial stiffness in CKD esp. in dialysis patients.
Author’s response:
cfPWV has been accepted as a standard tool for assessment of arterial stiffness [reference no. 33 and 34] and has shown by a recent study as an independent predictor of all-cause and cardio-vascular mortality in chronic HD patients [reference no. 34].
However, it still considered as a surrogate marker. Effects on hard outcomes such as cardiovascular event or morality should be explored in further long-term studies. We mentioned this issue in the discussion part [line 370-381]
- The observational period was relative short, 24 weeks. Thus, a significant finding of arterial stiffness in hemodialysis patients could be not significant changes.
Author’s response:
We agree that our observation period is relatively short and have mentioned it in the limitation [line 378-381]
- No circulating MK-7 levels in the study subjects were found in the text. Therefore, it is difficult to ascertain the compliance of MK-7 supplement in the study subjects.
Author’s response:
We did not measure circulating MK-7 levels in our study. However, There were still no established consensus levels for diagnosis and assessment of the degree of severity regarding to vitamin K deficiency by using the direct vitamin K levels or indirect biomarkers measurement of vitamin K status such as serum osteocalcin or serum dp-uc MGP levels. Therefore, we were unable to demonstrate the correlation between the improvement of vitamin K deficiency and the decreased vascular stiffness in ESKD patients. [mentioned in limitations line 381-388]
Patients’ drug compliance was reviewed by investigators and HD nurses from each HD centers every 4 weeks by counting returned pills. [mention in line 125-127] We also reported compliance in result section (subsection 3.5; Adverse event and compliance) [line 273-274]
Minor
- P-binder category, calcimimetic drugs(oral or iv injection)?
Author’s response:
We described phosphate binders category in Table 1 due to some participants were using multiple phosphate binders we show number of phosphate binder together with percentage of participants using calcium, aluminum, lanthanum, and sevelamer respectively.
All calcimimetics drug used were oral cinacalcet. (this information was added to the footnote of table 1)
- The participants had a well-controlled lipid profile, lipid-lowering agents were routinely prescribed in the HD subjects?
Author’s response:
Most participants using lipid-lowering agents prior to enrollment (70% in control and 74% in MK-7 group) In all centers, lipid-lowering agents were prescribed based on standard recommendation of using lipid lowering agent by KDIGO Clinical Practice Guideline for Lipid Management in CKD and local guideline by the Nephrology Society of Thailand.
All patients were advised to continue their regular medications, including antihyperten-sive drugs, phosphate binders, and lipid-lowering agents during the study period. [mentioned in Line 148-152]
- Circulating MK-7 levels was not sig. different between two groups at 24 weeks. How to explain? Is this one of reasons of non-significant change in arterial stiffness in the study subjects?
4. What was the circulating MK-7 levels in DM vs non-DM subjects?
Author’s response (for no 3,4):
We did not measure circulating MK-7 levels in our study.
(Already discussed above in the major comment no.4 and in the limitation section.)
Reviewer 3 Report
Dear authors and editor
This is an interesting article which showed vit K supplement might be beneficial for those who are under hemodialysis (HD) and with diabetes (DM)
with lower Carotid-femoral pulse wave velocity (cfPWV).
Several points need to be clarified before publishing.
1. Why use intention to treat (ITT) instead of those who complete the study?
2. The baseline prevelence of vit K deficiency among thailand may be mentioned with citation for better understanding of the backgriund.
3. Are those who are under new kind of thrombus prophylytic agent (factor X
inhibitor) exlcuded from the study? Why and why not?
4. Table 1 should label <intervention> and control group.
Als, the black bar can be removed if there is no special meaning to comply with the journal format setting.
5. Most important point: the control group is with higher BP( both SBP and DBP) with p=0.07, and higher prevelance of smoker, which both contributed to arterial stiffness.
If we control the factor of DM, the control group is very likely to have a stiffer artery in the first place.
Furthermore, is the number of both study and control groups in DM (14 vs 16) sufficient enough for reach the conclusion?
6. Another interesting finding in Table 2, those who are under HD three times a week seem to have a lower cfPWV in the control group (p=0.024)
compared with study group. Why?
I will suggest major revision first.
Minor issue:
The black bars in table 1 and 2 can be omitted.
The format setting may be adjusted. (ex: The large area of blank in page 5)
Moderate editing of English language is needed.
Author Response
Response to reviewer#3
This is an interesting article which showed vit K supplement might be beneficial for those who are under hemodialysis (HD) and with diabetes (DM) with lower Carotid-femoral pulse wave velocity (cfPWV). Several points need to be clarified before publishing.
- Why use intention to treat (ITT) instead of those who complete the study?
Author’s response: In order to preserve randomization and balance both known and unknown confounding factors we choose the ITT principle for main analysis in this study.
Due to the reviewer’s comment, we did a post-hoc sensitivity analysis in patients who completed the study which showed the same results. (Added new supplementary table S1)
- The baseline prevelence of vit K deficiency among thailand may be mentioned with citation for better understanding of the backgriund.
Author’s response:
There were no previous investigation of prevalence of vitamin K deficiency among Thai hemodialysis patients. However, data from other countries were mentioned in the introduction section [line 72-79]. We also had the data in Thai CKD patients (ref no.14) and demonstrated low vitamin K level (demonstrated by low serum dp-uc-MGP) had significant with vascular calcification and vascular stiffness (Reference no. 11) which had added to line 76-78..
- Are those who are under new kind of thrombus prophylytic agent (factor X inhibitor) exlcuded from the study? Why and why not?
Author’s response:
- We excluded only patient on warfarin due to potential drug interaction and interfere of the anticoagulant effect when prescribing vitamin K.
- Currently, there were no dialysis patients in all participating centers using direct oral anticoagulant. Unlike warfarin, their mechanism of action does not involve vitamin K, it might be no drug-drug interaction with MK-7, however, further studies are still needed.
- Table 1 should label <intervention> and control group.
Als, the black bar can be removed if there is no special meaning to comply with the journal format setting.
Author’s response:
Thank you for your suggestion, we have changed the label and removed the black bar from all tables.
- Most important point: the control group is with higher BP(both SBP and DBP) with p=0.07, and higher prevelance of smoker, which both contributed to arterial stiffness. If we control the factor of DM, the control group is very likely to have a stiffer artery in the first place.
Author’s response:
We agree with the reviewer that smoking and blood pressure could affects the outcome so, we had performed a sensitivity analyses by include smoking status, SBP and DBP in to the main model as adjusting factors which showed no difference from the main analysis (added supplementary table S1 and Line 212-215 in the main text)
Furthermore, is the number of both study and control groups in DM (14 vs 16) sufficient enough for reach the conclusion?
Author’s response:
We agree that the findings in the subgroup analyses should be interpret with caution due to multiple analysis issue which type I error could be increased. It should be mentioned as hypothesis generating and further study in this population is suggested. (mentioned in Line 390-393)
- Another interesting finding in Table 2, those who are under HD three times a week seem to have a lower cfPWV in the control group (p=0.024) compared with study group. Why?
Author’s response:
We think it could be explained by volume status which may be more adequately controlled in thrice-weekly HD patients compared to twice weekly HD. (mentioned in discussion section, line 346-354)
Minor issue:
The black bars in table 1 and 2 can be omitted.
The format setting may be adjusted. (ex: The large area of blank in page 5)
Author’s response:
Thank you very much, the format were modified as suggested.
Round 2
Reviewer 2 Report
The authors have addressed previous concerns from my critiques.
Author Response
Thank you for your helpful suggestions which improved the revised version of this manuscript.
Reviewer 3 Report
thanks for the reply, no further questions now.
Native english speaker revision may improve the fluency of this article.
Author Response
Thank you for your helpful suggestions which improved the revised version of this manuscript. We have consulted our professional native English editing service provided by our institution. Please see the English editing certification letter attached.
